# Interplay between Artificial Intelligence and Biomechanics Modeling in the Cardiovascular Disease Prediction

**DOI:** 10.3390/biomedicines10092157

**Published:** 2022-09-01

**Authors:** Xiaoyin Li, Xiao Liu, Xiaoyan Deng, Yubo Fan

**Affiliations:** 1Beijing Advanced Innovation Centre for Biomedical Engineering, Key Laboratory for Biomechanics and Mechanobiology of Chinese Education Ministry, School of Biological Science and Medical Engineering, Beihang University, Beijing 100083, China; 2School of Engineering Medicine, Beihang University, Beijing 100083, China

**Keywords:** artificial intelligence, cardiovascular diseases, machine learning, cardiovascular biomechanics modeling

## Abstract

Cardiovascular disease (CVD) is the most common cause of morbidity and mortality worldwide, and early accurate diagnosis is the key point for improving and optimizing the prognosis of CVD. Recent progress in artificial intelligence (AI), especially machine learning (ML) technology, makes it possible to predict CVD. In this review, we first briefly introduced the overview development of artificial intelligence. Then we summarized some ML applications in cardiovascular diseases, including ML−based models to directly predict CVD based on risk factors or medical imaging findings and the ML−based hemodynamics with vascular geometries, equations, and methods for indirect assessment of CVD. We also discussed case studies where ML could be used as the surrogate for computational fluid dynamics in data−driven models and physics−driven models. ML models could be a surrogate for computational fluid dynamics, accelerate the process of disease prediction, and reduce manual intervention. Lastly, we briefly summarized the research difficulties and prospected the future development of AI technology in cardiovascular diseases.

## 1. Introduction

Cardiovascular disease (CVD) is an important cause that threatens human health and represents a heavy economic burden on society and families [1,2]. Etiological investigations have found that the occurrence of cardiovascular disease involves a variety of risk factors [3,4,5], including high cholesterol, hypertension, diabetes, age, gender, genetics, unhealthy diet, obesity, smoking, alcohol, lack of exercise and environmental factors, as shown in Figure 1. Although managing these risk factors contributes to the control of cardiovascular disease, it remains the most common cause of morbidity and mortality worldwide [6,7]. Therefore, early, accurate diagnosis and prognosis assessments are the key point for improving and optimizing the prognosis of CVD [8].

Artificial intelligence (AI) is an exciting new field in cardiovascular disease, revolutionizing medical practice [9]. AI can effectively assist physicians in diagnosing cardiovascular diseases and conducting continuous monitoring so as to achieve early detection and treatment, thus reducing the occurrence of acute cardiovascular events and improving the prognosis [10]. AI has quite a few successful applications, such as imaging and pathological diagnosis [11], drug design and discovery [12], health management [13], disease prediction [14], medical rehabilitation [15], and laboratory medicine [16]. Advanced imaging and testing techniques have accumulated vast amounts of raw data, which are the basis of AI exploration. Fully use of artificial intelligence can revolutionize the current diagnosis and treatment model due to the complex and changeable structure of the cardiovascular system and play an important role in the prevention of cardiovascular diseases.

In this review, we systematically summarized the recent research progress of artificial intelligence applications for cardiovascular diseases, including the prediction of morbidity or mortality of the cardiovascular disease and the prediction of cardiovascular biomechanics modeling. We focused on machine learning−based models to predict CVD and ML−based vascular hemodynamic geometries, equations, and methods. Finally, we briefly summarized several common difficulties of AI technology and prospected the future development of AI technology in cardiovascular disease.

## 2. Overview of Artificial Intelligence

Artificial intelligence can be referred to as the science of realizing human intelligence on machines (Computers). In the past 70 years, artificial intelligence has been widely applied to many subjects and achieved fruitful results [17]. Machine learning (ML) is a subset of artificial intelligence, which refers to algorithms and statistical models that machines can learn independently, learn hidden patterns in data, and make accurate predictions to improve the performance of specific tasks [18]. According to how the data is learned, machine learning is mainly divided into reinforcement learning, supervised learning, semi−supervised learning, and unsupervised learning [19].

Since the 2010′s, ML algorithms have continuously improved, becoming more sophisticated and containing hierarchies, which gave rise to the popular Deep Learning (DL). Deep learning, a sub−field of machine learning, has been extensively studied and imitates the working model of the human brain and uses a multi−layer neural network to train data and develop an automatic prediction model (Figure 2) [20,21,22]. The DL−based model could automatically extract meaningful features from data on multiple levels. At the same time, the algorithm requires a certain degree of judgment by analysts in ML approaches to defining relevant features, such as feature selection, and it is popular in complex physical modeling, especially in nonlinear and high−dimensional functions [23,24]. Today, DL models have reached some important milestones, and various frameworks are emerging, such as the artificial neural network (ANN) [25], the deep neural network (DNN) [26], the convolutional neural network (CNN) [27] and the physics−informed neural network (PINN) [28], and so on.

Machine learning and deep learning algorithms have been used to accelerate the diagnosis and prediction of cardiovascular diseases [27,29,30,31]. For more detailed examples of the performance of ML and DL approaches for cardiovascular applications, we refer to Section 3.

## 3. Application of Artificial Intelligence in the Prediction of Cardiovascular Disease

We mainly discuss two types of ML−based approaches to predict cardiovascular diseases. The first approach is to build a machine learning model that directly outputs the incidence, mortality, or prognosis of CVDs by inputting clinical follow−up data and risk factors of subjects without CVDs, or clinical data and medical imaging of patients with CVDs. However, it requires massive patient data for training, and conventional prediction models are inadequate for assessing disease progression in complex lesions owing to the patient−specific anatomical, physiological, and functional difficulties. Another approach is to predict the complex patient−specific pathological process of CVDs by learning cardiovascular biomechanics based on numerical simulation. Computational models based on physical principles of cardiovascular systems, combined with medical imaging and patient characteristics, enable the derivation of hemodynamic information (e.g., velocity, pressure, and stress) inaccessible by medical images alone. Machine learning could be applied to the basic model and governing equations of the cardiovascular system and new numerical methods to accelerate the simulation process and realize personalized flow simulation. Both approaches are of great significance in evaluating the occurrence of cardiovascular diseases (Figure 3).

### 3.1. Prediction of Cardiovascular Morbidity or Mortality

Cardiovascular disease has long incubation periods and complicated pathogenic factors, which leads to the failure of timely identification and treatment [8]. A risk assessment system for CVDs could be established by mining the quantitative relationship between various related risk factors and their influence on the results. The most widely used model for cardiovascular diseases is the Framingham risk score (FRS) [32]. It predicted the occurrence probability of cardiovascular and cerebrovascular diseases in the next ten years according to the cholesterol level and non−cholesterol level factors and evaluated the risk by scoring the corresponding indicators. In addition to the FRS, the commonly adopted conventional risk prediction models are the systematic coronary risk evaluation score (SCORE) [33] and the atherosclerosis cardiovascular disease (ASCVD) [34]. Different predictive indicators were constructed through their research methods to study the risk factors of CVDs.

To improve the accuracy and speed of disease diagnosis of the above prediction models, AI−based approaches were applied to CVDs, which would help doctors to identify patients with different risk layers in advance and further reduce the incidence of mortality and adverse events. Several studies compared the ML−based model with the traditional risk prediction model (Table 1). For example, 13−year follow−up data from 6459 participants without cardiovascular disease were used to construct a machine learning model to calculate CVDs risk, which believed that the machine learning risk calculator could significantly improve risk stratification and reduce adverse events, resulting in the sensitivity of 0.86, specificity of 0.95, and the area under the curve (AUC) of 0.92 [35]. Data from the prospective study of 2020 adults trained in three ML models obtained similar results, in which random forest gave the best results [36]. Additionally, Alaa et al. [37] developed a DL−based prediction model with 473 reference variables in each case from 423,604 residents without cardiovascular disease, and their AutoPrognosis model improved risk prediction (AUC: 0.774) compared to the FRS (AUC: 0.724) and COX PH mode (AUC: 0.758), which proved that the automatic prediction model has better performance.

Artificial intelligence is also utilized in prognosticating cardiovascular outcomes based on imaging data combined with clinically available risk predictors, effectively reducing complications and sudden death events. Motwani et al. [38] used a regression model based on an iterative LogitBoost algorithm for mortality prognostication of 10,030 patients with suspected coronary artery diseases (CADs) who underwent coronary computed tomography angiography (CCTA) imaging and 5−year followup, and their performance (AUC: 0.79) was better compared to the FRS or CCTA severity scores alone. Another study [39] obtained similar results by using the ML−based model to predict the prognosis of 8844 patients with complete CCTA risk score information and at least 3−year follow−up for myocardial infarction and death, resulting in an AUC of 0.771. Coronary artery calcification (CAC) is one of the independent predictors of cardiovascular events, and machine learning has been combined with coronary artery calcification measurement to realize automatic identification [40]. Wolterink et al. [48] used a CNN to automatically identify and quantify CAC in CCTA images of 50 patients, which eliminated the need for coronary artery extraction and was expected to reduce unnecessary radiation doses in the future. Determining the degree of coronary artery stenosis is particularly important for patients with CADs, which determines the next treatment plan for patients. Detection and quantification of coronary artery stenosis is probably the most important clinical application of CCTA. Kelm et al. [49] used the ML algorithm to automatically identify and classify coronary artery stenosis caused by calcified and non−calcified plaques. Similarly, Zreik et al. [41] used multi−scale CNN to automatically identify functionally significant coronary artery stenosis in CCTA images of 166 patients. The results suggested that functional coronary artery stenosis could be determined by automatic analysis of myocardium in resting CCTA images without observation of the patient’s coronary artery anatomy, which might reduce unnecessary invasive fractional flow reserve (FFR) examination in the future. Besides CCTA, other imaging techniques combined with artificial intelligence methods have also been applied to predict CADs. Myocardial perfusion imaging showed [27] that the DL algorithm could predict the occurrence of adverse events more accurately and with higher accuracy than traditional prediction models. In addition, a CNN−based plaque detection system was employed to learn plaque classification directly from intravascular optical coherence tomography (IVOCT), resulting in an accuracy of 0.917, sensitivity of 0.909, and specificity of 0.924. Their results demonstrated that it was feasible to establish a plaque detection system based on deep learning [42].

Artificial intelligence also has many applications for other cardiovascular diseases besides CADs. Based on echocardiographic data, ML algorithms were used to establish the discrimination models between hypertrophic cardiomyopathy from physiological hypertrophy seen in athletes [43] and between hypertrophic cardiomyopathy and constrictive pericarditis [44]. Their ML−based models had higher diagnostic sensitivity and specificity. Diller et al. used a Deep learning algorithm to categorize diagnosis, disease complexity, and NYHA class in adult congenital heart disease or pulmonary hypertension through an 8−year follow−up of 10,019 adult patients, with an accuracy of 0.911, 0.97, and 0.906, respectively [31]. Additionally, the heart sound−based detection methods for heart failure were proposed through machine learning and end−to−end deep learning [45]. Based on the records of 947 subjects, 15 repeatable machine learning models were identified to distinguish the different stages of chronic heart failure with an accuracy of 0.929. This approach made it easier to identify patients with heart failure, which has the potential of a home chronic heart failure monitor. Artificial neural networks (ANNs) could also identify stroke and stroke−like diseases intelligently by analyzing a large amount of data [25]. The application of ML in predicting postoperative mortality and rehospitalization after surgery would also be of great significance. Data from 11,709 patients undergoing percutaneous coronary intervention verified that machine learning was more predictive in identifying postoperative mortality 180d after surgery and rehospitalization for chronic heart failure 30 d after surgery [46]. Another study [47] also showed that the ML model (AUC: 0.795) was more accurate in predicting mortality after elective cardiac surgery than EuroSCORE II (AUC: 0.737) or the logistic regression model (AUC: 0.742). Compared with the traditional prediction score and prediction model, the ML−based model would be faster and more accurate, which could alter the prediction method of cardiovascular disease and improve the prediction accuracy.

### 3.2. Prediction of Cardiovascular Biomechanics Modeling

Cardiovascular biomechanics could also be applied to predict the occurrence of cardiovascular diseases [50,51] as the critical roles of blood flow and arterial wall mechanics and their interactions in the function of the human cardiovascular system [52,53,54,55]. Therefore, from the perspective of cardiovascular biomechanics, it is feasible to diagnose or prevent cardiovascular diseases indirectly through hemodynamic parameters, such as velocity [56,57], pressure [56,58], and wall shear stress (WSS) [59,60,61,62]. It is well documented that the flow in blood vessels is very complicated, and it is difficult to predict directly [63]. The development of vascular fluid dynamics has depended on basic geometries, equations, and computational methods [64,65,66]. Based on artificial intelligence technology, the basic model and governing equations of the fluid system and new numerical methods could be developed to accelerate the process and reduce manual intervention.

#### 3.2.1. Traditional Computational Modeling and Simulation

Computational modeling and simulation methods were frequently applied to solve flow problems in blood vessels. Numerical analysis methods of cardiovascular biomechanics mainly rely on a grid−based approach, including finite difference analysis (FDA) [67], finite volume analysis (FVA) [68], or finite element analysis (FEA) [69,70]. Combined with medical images, it is very effective for cardiovascular function analysis in solving the basic physical equations of the flow field with high precision in discrete form and studying the fluid motion and its interaction with other media [71,72,73,74,75,76]. The current workflow for patient−specific computational modeling and simulation applications mainly consists of three steps [77]: (1) the vascular anatomic geometry of the patient is obtained from clinical image data, mainly through manual labeling; (2) the specifying constitutive relation of material properties, boundaries, and hemodynamic loading conditions are set up in the computational model; (3) the computational model is submitted to an appropriate numerical solver to obtain the simulation results. The computational modeling needs a lengthy model setup and long computing time to complete the analysis of one single patient [78,79]. Therefore, the computational fluid dynamics (CFD) method cannot be applied for large queues of patients or time−sensitive clinical applications requiring rapid feedback to clinicians, such as percutaneous coronary intervention [46]. Although the present model reduction methods, such as dynamic modal decomposition (DMD) [80] and proper orthogonal decomposition (POD) [81], greatly reduce the solution of complex systems and improve the efficiency of modeling and solving, the traditional model reduction methods are still difficult to be applied to multi−scale, transient and discontinuous processes [82,83].

Machine learning could break down computing tasks and make all kinds of machine assistance possible in cardiovascular biomechanics modeling. In particular, the ML approach could be applied to the extraction of geometric features, the study of governing equations, and the surrogate for CFD [84,85,86]. In addition, replacing certain finite element components with machine learning models can achieve faster computation time, especially for multi−scale problems that require nested finite element simulations [87,88,89].

#### 3.2.2. ML−Based Hemodynamics with Vascular Geometries, Equations and Methods

##### Geometric Modeling

The traditional method of computational fluid dynamics needs to extract structures and features of blood vessels manually [77,78,79], which is labor and time−consuming. In recent years, artificial intelligence technology was proposed to automatically and quickly extract geometric features as input in the computational model. For example, the nearest distance was used between each grid point in the rectangular grid and the boundary as input to predict the resistance coefficient, which achieved good results in simple two−dimensional geometric shapes such as circles [90]. In addition, there were related studies using features extracted by deep learning technology [91,92], and the compressed expression of geometric shapes of autoencoders was put into neural networks. Other studies represented the geometry by image pixels [93], which are discretized into a 2D/3D image.

ML models have been used to automatically segment medical images for creating 3D computer models in recent years. Especially in patient−specific biomechanical modeling, each cardiovascular disease leads to multi−feature 3D morphologies, such as atherosclerosis [94], aneurysm [95], and occlusive diseases [96,97]. For instance, Berhane et al. [98] used deep learning to generate an automatic 3D segmentation model of the aorta based on 4D−flow magnetic resonance imaging (MRI). The feasibility of using ML techniques and deformable methods for automatic geometry reconstruction and modeling of human organs from 3D medical images has been proved [99,100,101]. Liang et al. [85,102] used ML algorithms for automatic geometry modeling of aortic aneurysms from 3D medical images. Luo et al. [103] developed ML classifiers to infer the strength of ascending thoracic aneurysm from elastic properties. Zheng et al. [104] reduced the complexity of computerized tomography (CT) data for carotid artery bifurcation detection. Moeskops et al. [105] trained a single CNN model to segment the coronary arteries in cardiac computed tomography angiography. The complex shapes of atherosclerosis are mainly derived from vascular imaging [106], including intravascular ultrasound (IVUS) [107,108], angioplasty [109], MRI [98], and IVOCT [110]. ML−based approaches have been used for the analysis of imaging data to characterize plaque morphology. Iyer et al. [111] designed a CNN model for vessel segmentation in X−ray angiography images. To visualize the severity of coronary artery stenosis, Lee et al. [112] constructed a CNN−based fully−automated semantic segmentation model of coronary plaque in IVOCT images, resulting in high sensitivity and specificity classification of lipid and calcified plaque (the sensitivity/specificity were 87.4%/89.5% and 85.1%/94.2%, respectively). In addition, Tang et al. [113] proposed a deep neural network based on multi−scale features for automatic lumen segmentation for IVOCT images. Athanasiou et al. [114] used the ML−based OCT image segmentation to identify areas of atherosclerotic plaque. Abdolmanafifi et al. [115] compared the image segmentation accuracy of three models, and they demonstrated that the convolutional neural network (CNN) was very effective when applied as a feature extractor. In addition, ML was also widely used in cardiac image segmentation, including congenital heart disease [116] and whole−heart [117]. A fully automated approach has been developed for segmenting the mitral leaflets from 3D transesophageal echocardiography image data to facilitate visual and quantitative image analysis [118] and aortic valve modeling [119,120]. Oktay et al. [121] confirmed that CNN could accurately segment the left ventricle and depict anatomical morphological changes related to cardiac pathology. Other studies obtained similar results [122,123]. At present, the ML segmentation models cannot fully reach an agreement with anatomical structures or experts [124]. It is difficult to build ML−based models that can appropriately represent all morphologies due to the complexity of patient−specific shapes, and feature extraction with wide generalization remains a challenge (Table 2).

##### Governing Equation (ML−Based Partial Differential Equation)

The governing equation of the flow in blood vessels is generally regarded as Navier−Stokes (N−S) equation, which is a highly nonlinear partial differential equation (PDE) system [130], and the solution of the equation is always a difficult problem because of its complicated process. Nowadays, the machine learning approach is explored to assist solutions from the perspective of partial differential equations.

The sparse regression technology was used to learn the coefficients and derivative forms of each order in the Taylor series. This method implemented interpretable machine learning techniques but requires the construction of function libraries to ensure that the function involved is included [131,132]. Raissi et al. [87] learned coefficients and function terms in the Taylor series by constructing two neural networks of different depths. Since the function items were represented by neural networks, this approach could only learn the abstract expression of the original function but did not affect its application for practical problems, such as the solution of the N−S equation. Additionally, the data−driven method with the data assimilation method was combined to identify PDEs, which broadened the application scope of PDEs identification. For the known structure of PDEs, machine learning methods based on the Gaussian process were introduced into linear differential equation systems [133] and nonlinear partial differential equation systems [134] to identify scalar coefficients in the equations. In addition, the multi−fidelity Gaussian process was also introduced to predict random fields [135]. However, these methods were only applicable to models with fewer data.

A more recent ML paradigm, a physics−informed neural network, has been proposed to identify scalar parameters in partial differential equations [28,136,137]. It took partial differential equations as regularization terms in the process of neural network fitting data, which avoided the inapplicability of traditional numerical differentiation to noisy data. Alternatively, some methods were mainly introduced into the network training process in the form of the loss function, which focused on data fitting rather than solving mathematical equations itself [126,138]. In terms of future work, quantifying the uncertainty associated with neural network predictions is the focus of research. For data−driven differential equations, how to combine the basic law of conservation of physics to ensure the conservation law is worthy of being studied further.

##### A ML−Based Surrogate for Computational Fluid Dynamics

The artificial intelligence−based solution method for computational fluid dynamics is driven by data or physical models. The data−driven model is based on existing model equations, which constantly update or optimize the original empirical coefficients [129]. The physics−driven model completely abandons the existing model equations and builds a specific input−output relationship through machine learning based on certain physical knowledge (Table 2) [86].

The neural networks based on data firstly obtain high fidelity data according to the existing flow field simulation or experimental methods and then construct the neural network mapping relationship to replace the original partial differential equation after learning the data based on machine learning technology, which could obtain the numerical solution to the flow field quickly and efficiently [86,103]. Moreover, various data assimilation methods have been developed to combine experimental data with computational hemodynamic data to improve data fidelity [139,140,141]. Artificial intelligence algorithms usually take biomechanics simulation results as training data to predict interested hemodynamic parameters. For instance, Jordanski et al. constructed three ML approaches (Gaussian conditional random fields, multilayer perceptron neural network, and multivariate linear regression) to predict the WSS distribution of abdominal aortic aneurysm and carotid bifurcation models, and the strong determination coefficient of CFD simulation was verified [125]. Another study used the decision tree to estimate FFR from 34 pre−defined coronary arteries [142]. Similarly, other ML models were proposed to measure FFR from coronary computed tomography angioplasty, in which ML models were trained on a synthetic FFR dataset obtained from CFD simulation [143,144].

The successful development of ML models has been verified in massive applications. Recent work in machine learning combined with finite element calculations has also shown promise for collagen tissues [145].

The deep learning approach has gradually emerged in cardiovascular biomechanics modeling with the rapid development of artificial intelligence. For example, DL models were developed to directly evaluate the stress distribution of thoracic aortic aneurysms bypassing the finite element calculation process [102]. Liang et al. [77] demonstrated that DL models could quickly and accurately substitute for stress analysis, which is shown in Figure 4. Recent studies using ML models to predict blood flow velocity vector fields have obtained similar results [58], which is shown in Figure 5. Subsequently, the study demonstrated the feasibility of using a deep neural network as a fast and accurate surrogate for computational fluid dynamics to estimate the hemodynamics of the human arteries [57]. In addition, the convolutional neural networks have been wildly constructed as finite element alternatives to fluid dynamics analysis, which could compress the simulated state size and learn dynamics [91,146]. For instance, CNN could be used to reconstruct the high−resolution turbulent field without solving the governing equation. The input of the network was the low−resolution flow field pooled from high−resolution flow field images obtained from the direct numerical simulation method (DNS), and the method was validated in laminar cylindrical flow and isotropic turbulence [147]. Similarly, CNN also could be used to perform the parameter estimation in cardiovascular hemodynamics [128]. Five different neural networks were used to predict arterial wall stress in atherosclerotic patients, which also proved the superiority of convolution networks [127].

As a general−purpose function approximation [148], the AI−based model only approximates the complex nonlinear relationship between the input and output variables of the system. Given input parameters such as initial/boundary/operating conditions, parameters of interest such as velocity, pressure, and shear stress are obtained without traditional CFD simulations. For example, a physics−informed neural network has been proposed recently [149,150], in which the governing physical equation, such as Navier−Stokes equations, could be integrated into the neural network frameworks where the physical variables of interest could be expressed as functions of space and time. Raissi et al. [126,151] developed a physics−informed deep−learning framework to solve partial differential equations, and the neural networks were trained in several observed values of a specific flow field. With the governing equation as the constraint, the flow field prediction in a specific region could be realized without introducing boundary conditions, which is shown in Figure 6. Similarly, the conservative PINN method could be used to solve the flow field in multiple different sub−regions, which improved the applicability of the original method for solving complex boundary flow field problems [152]. The physics−informed neural network, which uses sparse measurement data to solve uncertain problems and simultaneously identify unknown parameters, has gained much attention in cardiovascular modeling [129,153]. In addition, PINN could be applied to improve the WSS quantification in blood flow problems where the inlet and outlet boundary conditions were not known but instead by assimilating a few measurement points [154].

The data−driven neural network requires quantities of high−fidelity data, but it is difficult to apply it when the sample data is small or cannot be obtained [155]. In the context of surrogate modeling, if boundary conditions are properly applied, the physical−driven deep learning model without labeled data could be developed to directly integrate physical equations and boundary conditions into the loss function of the neural networks. For instance, a physics−constrained, deep learning model could be developed to solve the high−dimensional stochastic elliptic partial differential equations based on the sample−free data method [156]. Similarly, CNN models also could be used in solving the partial differential equation with numerical methods. The image data of the flow field was put into networks as the initial solution instead of training sample data, and their result was better than data−driven networks [157,158]. Recent studies demonstrated that DL models also had great promise in solving high−dimensional nonlinear uncertainty quantification (UQ) problems [159]. PINN could find flow velocity, stress or pressure fields satisfying N−S equations at any specified point in the domain without any training data, which can be used as a substitute for traditional CFD. For example, Sun et al. [86] used a physics−constrained neural network−based alternative model to solve the parameterized N−S equations and simulated the ideal vascular flow without using any labeled training data (Figure 6). The flexibility provided by PINN makes it possible to solve complex cardiovascular biomechanical problems. In addition, PINN could also be used in varieties of complex flow fields [160,161,162].

Both data−driven models and physical−driven models have promoted the development of computational fluid dynamics and explored a new way for cardiovascular disease prediction. However, the success has only been demonstrated on several canonical issues [86,126,127], and further research is needed on the broader impact of complex cardiovascular disease. Therefore, the prediction and modeling techniques of cardiovascular fluid mechanics should be problem−oriented, and a broadly applicable ML model may require more effort in the future.

## 4. Challenges and Future Prospects

The increasing maturity of artificial intelligence and the continuous expansion of its application in the medical field brought revolutionary changes to medical practice. ML−based models could predict the morbidity or mortality of CVDs more accurately and faster than the traditional prediction model, which could alter the prediction method of CVDs. ML models could also provide new theories, methods, and research paradigms for biomechanics modelling. However, there are still several challenges that need ironing out. (1) Database: The early prediction of CVDs is inseparable from massive datasets and data quality. The high−precision mechanical calculation needs huge computing resources, and it is difficult to obtain data on some rare diseases [163]. In addition, the applications of medical data also involve patient privacy and ethical issues [164]. Therefore, it is necessary to obtain more accurate, authentic, and appropriate datasets and train and improve the algorithm on small samples [165]. High−quality data and appropriate storage methods are critical to the development of AI technology. (2) Validity and stability: Studies demonstrated that AI−based models usually have better prediction performance compared with traditional prediction models [38]. To make the models have clinical practical value, further testing with massive pathological datasets is significant to verify the effectiveness and stability of the proposed method [166], and the results from single−center also need to be verified by the multi−center and massive cases. (3) Generalization: Numerous ML models have been successfully used to predict CVDs in recent years [57,58,102,127]. However, it remains a challenge to better generalize the AI−based prediction models to future data for specific patients. For example, in patient−specific biomechanics applications, it is hard to develop ML−based models that can appropriately represent all complex 3D morphologies of each cardiovascular disease. Thus, a broadly applicable ML−based prediction model might require learning vast physiologically possible datasets. (4) Explainability: Artificial intelligence technology has high predictive performance in the medical field, but it is still hard to explain the decision−making process clearly. Machine learning has always been used as a “black box” in the research process, which leads to the inexplicability and uncertainty of the model [167,168]. Moreover, various subjective factors and the complexity of models affect the design and evaluation of AI models. The explainable AI model has received sustained attention in recent years [169], but the results were not perfect, and a universal system and unified evaluation index are lacking. In addition, models are valued more for accuracy than explainability. A research method with reproducibility and standard must be found in the future to compare and evaluate the explainability of the AI-based prediction model.

Given the above, artificial intelligence has made remarkable achievements in the past decades, but the data quality, as well as the validity, stability, generalization, and explainability of the model need to be further improved. AI will be a development direction and trend of medical treatment in the future, and it will alleviate the pressure of medical treatment in some aspects, improve the speed and quality of medical services, and promote the continuous development and progress of human medicine.

## 5. Conclusions

We summarized the recent advances of the machine learning−based model to directly predict CVDs based on risk factors or medical imaging findings, as well as the machine learning−based hemodynamics with vascular geometries, equations, and methods for indirect assessment of cardiovascular diseases. Machine learning models could be a surrogate for computational fluid dynamics, accelerate the process of disease prediction and reduce manual intervention.

## Figures and Tables

**Figure 1 biomedicines-10-02157-f001:**
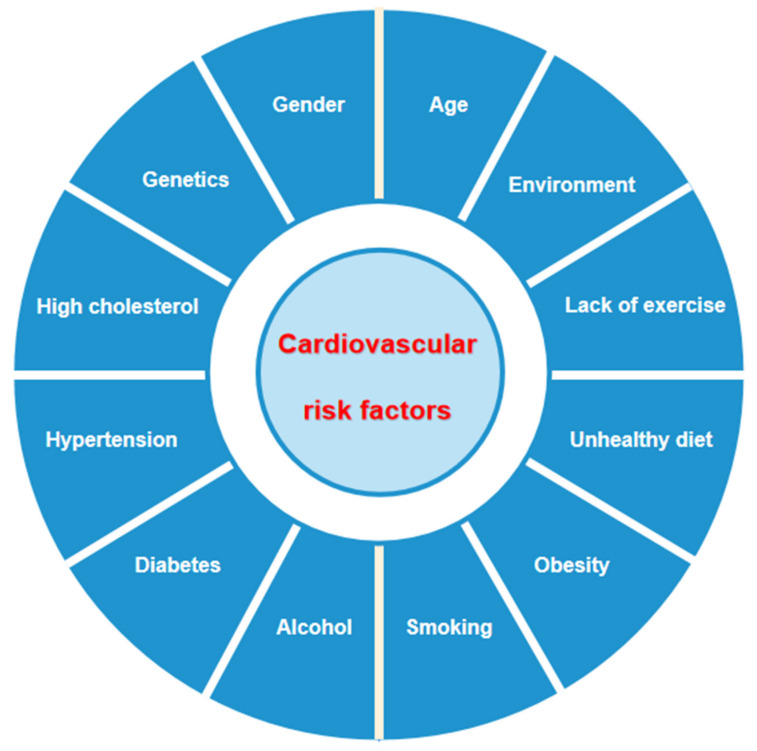
Risk factors associated with cardiovascular disease.

**Figure 2 biomedicines-10-02157-f002:**
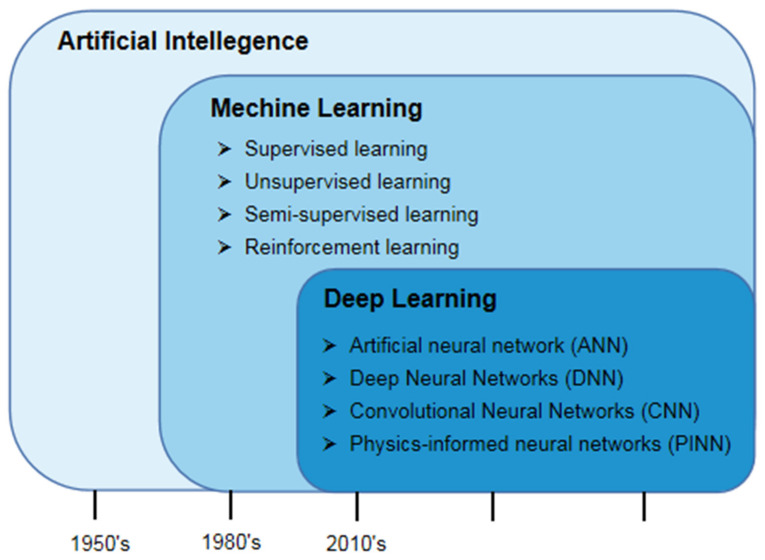
The overview development of artificial intelligence and the relationship between artificial intelligence, machine learning, and deep learning.

**Figure 3 biomedicines-10-02157-f003:**
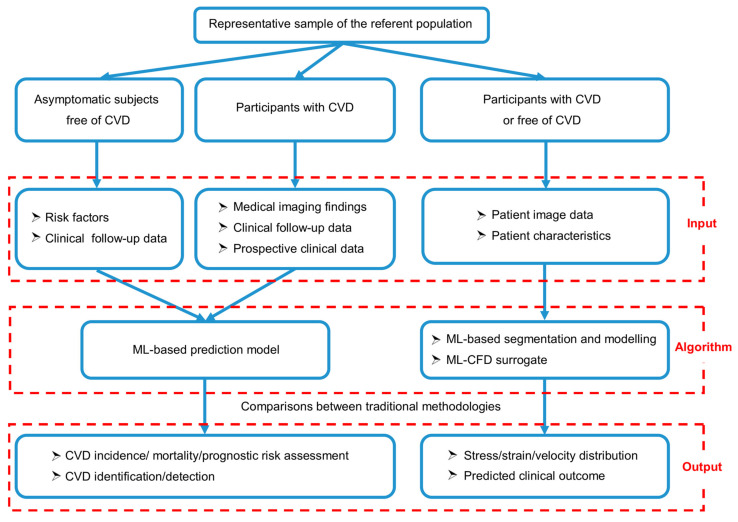
Two types of machine learning−based approaches were applied to evaluate cardiovascular disease risk.

**Figure 4 biomedicines-10-02157-f004:**
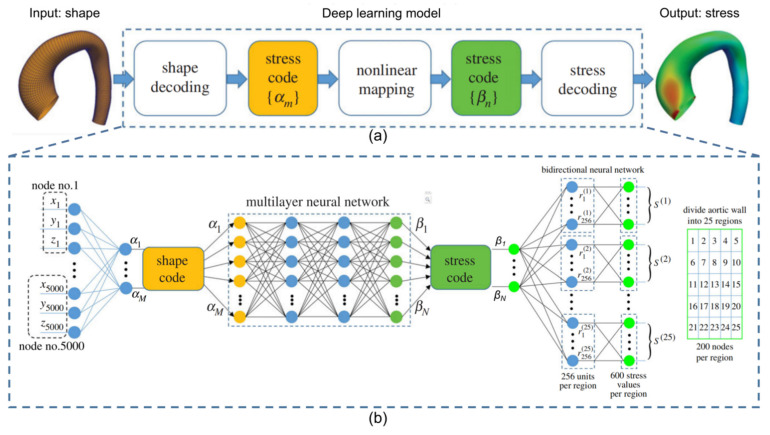
Case study using the deep learning (DL) technique as a surrogate of finite−element analysis for stress analysis. (**a**) The overall structure of the DL model, in which the input is an aorta shape and the output is the stress distribution of the artery wall. (**b**) The neural network for the shape encoding, mapping the shape code to the stress code, and the stress decoding and encoding. Panel (**a**,**b**) are adapted with permission from Reference [77], Journal of the Royal Society Interface.

**Figure 5 biomedicines-10-02157-f005:**
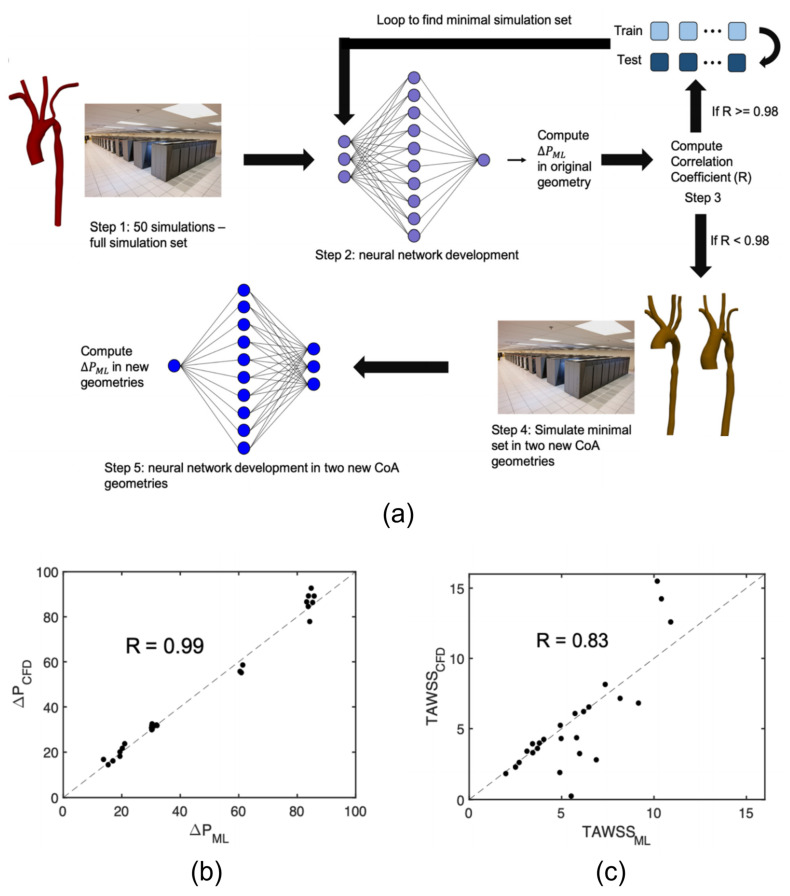
Case study for machine learning framework for surrogate modeling of pressure gradient and WSS for patients with coarctation of the aorta [58] (**a**) Design of experiments workflow. (**b**) ML results comparing predicted and simulated pressure. (**c**) ML results comparing predicted and simulated TAWSS.

**Figure 6 biomedicines-10-02157-f006:**
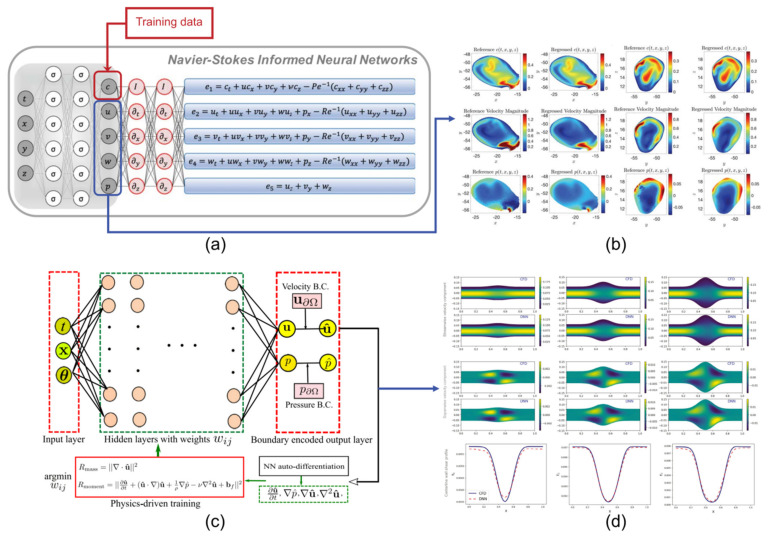
Case study for the physics−constrained, data−driven framework and data−free framework for surrogate modeling of fluid flows in aneurysms. (**a**) The data−driven structure of the Navier−Stokes−informed neural networks. The input data c is generated by using a direct numerical simulation, and the output is the quantitative hemodynamic parameters. (**b**) Contours of reference and regressed concentration, velocity, and pressure fields within the 3D intracranial aneurysm sac. (**c**) The data−free neural network is developed by coding boundary conditions with construction and trained by minimizing the loss function based on equations instead of CFD simulation data. (**d**) Contours of physics−constrained deep neural network predictions and CFD solutions of three different aneurysm geometries. Panel (**a**,**b**) are adapted with permission from Reference [126]. Science. Panel (**c**,**d**) are adapted with permission from Reference [86]. Elsevier.

**Table 1 biomedicines-10-02157-t001:** Machine learning−based prediction models of cardiovascular diseases.

Focus	Algorithm	Data Size	Input Variables	Performance (AUC)	Significant Discovery
CVD risk prediction [35]	ML (SVM)	6459	clinical data	0.92	ML algorithms significantly improved risk stratification while reducing adverse events.
CVD risk prediction [36]	ML (KNN, RF and DT)	2020	clinical data	Accu.: 0.83 Sens.: 0.89 Spec.: 0.46	The RF gave the best results, while the k−NN gave the poorest results.
CVD risk prediction [37]	AutoPrognosis (SVM, RF, kNN, AdaBoost and GBM)	423,604	clinical data	0.774	ML model had better efficiency than traditional risk calculators.
CAD mortality prediction [38]	ML (LogitBoost)	10,030	clinical and CCTA data	0.79	The accuracy of the ML model was better compared to the traditional or CCTA severity scores alone.
CAD risk prediction [39]	ML (XGBoost)	8844	clinical and CCTA data	0.771	The risk score based on ML had greater prognostic accuracy than current CCTA integrated risk scores.
CAC identification [40]	CNN + RF	50	CCTA data	/	CAC could be automatically identified and classified in CCTA using CNN and RF algorithms.
Coronary artery stenosis identification [41]	CNN + CAE + SVM	166	FFR and CCTA data	0.74	The CNN could be used to automatically identify functionally significant coronary artery stenosis.
Obstructive disease prediction [27]	DL	1638	MPI data	0.80/0.76	The DL algorithm could automatically interpret MPI more accurately.
CHD Plaque detection [42]	CNN	49	IVOCT data	Accu.: 0.917 Sens.: 0.909 Spec.: 0.924	It’s feasible to construct a DL−based clinical decision support system for plaque detection.
HCM discrimination [43]	ML (SVMs + RF) + ANN	139	STE data	0.795	The ML−based models had higher diagnostic sensitivity and specificity.
CP/ RCM discrimination [44]	ML (AMC, RF, SVM and kNN)	94	Clinical and STE data	0.962	The AMC gave the best results.
Prognosis prediction [31]	DL	10,019	Clinical and ECG data	Accu.: 0.906	It was feasible to build a DL−based model to estimate the prognosis in ACHD.
CHF identification [45]	ML (RF) + DL	947	Clinical and heart sounds data	0.893	The heart sound−based detection methods for different CHF phases were proposed through ML and DL.
ACI identification [25]	ANN	260	clinical data	Spec.: 0.862 Sens.: 0.8	ANN could be used for the recognition of ACI and differentiation of ACI from stroke intelligently.
Perioperative mortality prediction + Readmission [46]	ML (RF)	11,709	Perioperative clinical data	0.9/0.88	ML was more predictive in identifying postoperative mortality 180d after PCI and rehospitalization for CHF 30d after surgery.
Perioperative Mortality prediction [47]	ML (GBM, RF, Naïve Bayes, SVM)	6520	Perioperative clinical data	0.795	ML model was more accurate in predicting mortality after elective cardiac surgery than the traditional prediction model.

(AUC: Area under the curve; CVD: Cardiovascular disease; ML: Machine learning; SVM: Support vector machine; KNN: K−Nearest neighbor; RF: Random forests; DT: Decision tree; Accu.: Accuracy; Sens.: Sensitivity; Spec.: Specificity; GBM: Gradient boosting machines; CAD: Coronary artery disease; CCTA: Coronary computed tomography angiography; CAC: Coronary artery calcification; CNN: Convolutional neural network; CAE: Convolutional autoencoder; FFR: Fractional flow reserve; DL: Deep learning; MPI: Myocardial perfusion imaging; CHD: Coronary heart disease; IVOCT: Intravascular optical coherence tomography; HCM: Hypertrophic cardiomyopathy; ANN: Artificial neural network; STE: Speckle−tracking echocardiographic; CP: Constrictive pericarditis; RCM: Restrictive cardiomyopathy; AMC: Associative memory classifier; ECG: Electrocardiograph; ACHD: Adult congenital heart disease; CHF: Chronic heart failure; ACI: Acute cerebral ischemia; PCI: Percutaneous coronary intervention; GBM: Gradient boosting machines).

**Table 2 biomedicines-10-02157-t002:** Artificial intelligence−based surrogate for computational fluid dynamics.

Algorithms	Name of Authors	Objectives	Training Set	Significant Discovery
ML	Jordanski et al. [125]	WSS	FEA results	Three ML models (MLR, MLP, GCRF) were developed for the calculation of WSS distribution, and the GCRF achieved the highest coefficient of determination (0.930–0.948) for the AAA model and (0.946–0.954) for carotid bifurcation model.
ML	Feiger et al. [58]	Pressure, WSS	LBM results	The 3D simulation−based ML model was developed to accurately predict pressure gradient across the stenosis and WSS for patients with coarctation of the aorta.
DL	Li et al. [57]	Velocity, Pressure gradient	FEA results	The hemodynamic prediction results of deep learning was in agreement with the conventional CFD method, but the calculation time was reduced 600−fold.
DL	Raissi et al. [126]	Velocity, Pressure	DNS results	A physics−informed deep−learning framework was capable of encoding the Navier−Stokes equations into the neural networks while being agnostic to the geometry or the initial and boundary conditions.
DNN	Madani et al. [127]	Stress	FEA results	The DNNs outperformed alternative prediction models and performance scales with the amount of training data.
DNN	liang et al. [102]	Pressure, Velocity	FEA results	The trained DNNs were capable of predicting the steady−state distributions of pressure and flow velocity inside the thoracic aorta with an average error of 1.9608% and 1.4269%.
CNN	Kai et al. [128]	Velocity	DNS results	The CNN model was found to reconstruct turbulent flows from extremely coarse flow field images with remarkable accuracy.
PINN	Arzani et al. [129]	WSS	N−S equations	PINN was used to obtain near−wall hemodynamics and WSS data from sparse velocity measurements and without knowledge of the inlet/outlet boundary conditions.
FC−NN	Sun et al. [86]	Velocity, Pressure	N−S equations	A physics−constrained deep neural network−based approach was developed for surrogate modeling of fluid flows without relying on any simulation data.

(ML: Machine learning; WSS: Wall shear stress; FEA: Finite element analysis; MLR: Multivariate linear regression; MLP: Multilayer perceptron neural network; GCRF: Gaussian conditional random fields; AAA: Abdominal aortic aneurysm; LBM: Lattice Boltzmann method; DL: Deep learning; CFD: Computational fluid dynamics; DNS: Direct numerical simulation; DNN: Deep neural network; CNN: Convolutional neural network; PINN: Physics−informed neural network; N−S equations: Navier−Stokes equations; FC−NN: Fully−connected neural network).

## Data Availability

Not applicable.

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
