# Peer review of "Interplay between Artificial Intelligence and Biomechanics Modeling in the Cardiovascular Disease Prediction"

_biomedicines, 2022, doi:10.3390/biomedicines10092157_

Round 1

Reviewer 1 Report

In this review article, the authors provide an overview of artificial intelligence (AI) and its application in predicting cardiovascular disease (CVD). They focus on machine learning (ML), and summarize its various uses in these efforts, including in direct prediction models and indirect hemodynamic assessment. They illustrate the various roles of ML in the latter, including vascular geometric modeling, governing partial differential equation identification, and—most extensively—surrogate methods to replace computational fluid dynamics. Overall, the review is timely, addresses a topic of growing importance and promise, and cites and extensive body of literature. However, there are multiple opportunities for the authors to improve the clarity, focus, and organization of their article, and sections which can be made more comprehensive.

While the fit of the article subject matter is at the discretion of the editors, I note that the selected topic is “Blood Physiology: Molecular Mechanisms of Vascular Wall Functioning.” The article, while related, does not seem to directly address to this topic. The authors might consider how the article can be built upon to more directly address the topic, though this pertains to the journal focus and not the inherent quality of the article (so does not factor into my judgement of the paper or recommendation).

The degree of self-citation is modest and appropriate; those cited works are reasonably relevant to the article content.

There are several grammatical and spelling errors throughout the manuscript, and some acronyms (e.g., CAD) are not defined before use. Though the meaning of the text can generally still be readily understood, the authors (or editors) should carefully review and revise the text, as necessary.

In the abstract, the authors call their article a “tutorial review.” What does this mean? I typically associate tutorials with teaching practicable skills or techniques (like a “how to” guide), but this article does not do so.

Table 1 is incredibly important in reporting the key aspects of work reviewed in the article, but it is internally inconsistent, disorganized, and confusing in its current form. The “Data (Imaging)” column is particularly confusing and inconsistent. For example, for some studies, the data is reported as time to follow up, while for others it’s the type of data (e.g., type of imaging). The follow-up time and type of data are independent, so it’s unclear why they’re reported interchangeably under the same heading. (What type of data was available at baseline and follow-up for those studies for which time to follow-up is reported?) Additionally, “PCI” is listed as “Data (Imaging),” but PCI is a medical procedure, not a type of data. “Prospective clinical data” and “473 available variables” are incredibly vague and ambiguous descriptors. The “Objectives” also don’t all make sense. Why are diseases listed as objectives? (Note also that coronary artery calcification, listed as an “objective” as if it were a disease, is a sign of cardiovascular disease, but not a CVD itself.) How is “Mortality” an objective? (Does it mean prediction of mortality? If so, how does that differ from “risk prediction”?) What is being estimated, detected or identified? What risks are being predicted? Sometimes the algorithm is just listed as “ML,” while other list the specific type of machine learning algorithm. Some acronyms are defined but never used (e.g., AUC). This table is a core feature of your article. If this information is not reported in the manuscript text (it currently is not), then the table must be more precise. The text can elaborate on the details that you do not want to include in the table.

Table 1 covers methods and articles reviewed in Section 3.1. The authors should strongly consider adding a similar table (or extending Table 1) summarizing articles/methods reviewed in Section 3.2.

As a general comment, the authors are encouraged to review Section 3 (“Application of artificial intelligence in the prediction of cardiovascular disease”) and consider where additional context, detail, and descriptions would help readers to better understand the described works, their relevance, and their relationships.

The inclusion of highly tangential work, such as references to work in segmenting the prostate in MRI (section 3.2.2.1) and smoke simulation in computer graphics and animation (section 3.2.2.3), adds confusion and detracts from the focus of the paper. The authors should review the content and references to ensure they are all relevant and contribute meaningfully to the review.

Additional diagrams, particularly a schematic illustrating the framework you’re using to divide and categorize the work you’re reviewing, would be helpful. Additionally, more diagrams illustrating concepts described in the text (like that shown in Figure 4) would also be beneficial.

The authors inconsistently and variably refer to PINNS as “physics-informed neural networks” and “physical informed neural networks;” the former is the correct terminology.

In Section 3.2.2.1 (“Geometric modeling”), the authors note that “ML could also realize automatic image segmentation to create 3D computer models in cardiovascular biomechanics modeling.” While they cite several relevant examples, I believe a comprehensive review of the interplay between AI and biomechanics modeling in CVD is incomplete without a brief discussion of work to model atherosclerotic vessels or lesions using ML-derived morphologies. This is one of the key areas of research beyond hemodynamic simulation/modeling in which AI is being leveraged to conduct biomechanical modeling for the most prevalent CVDs, atherosclerosis and coronary artery disease (which the authors correctly note produces “multi-feature 3D morphologies” earlier in the paragraph). Earlier in the article, the authors cite Gessert et al. [44] (who developed a method to identify the presence of plaque in an image, though not plaque morphology/distribution within images). There is a wider body of literature describing more advanced methods which yield lesion morphology, typically derived from intravascular imaging (intravascular optical coherence tomography or intravascular ultrasound), that can be implemented to conduct stress analysis or even mechanical characterization.

The authors indicate in the abstract that they “briefly prospected the research difficulties and prospected the future development of AI technology in cardiovascular diseases,” but it is unclear what content of the manuscript actually fulfills this purpose. A dedicated section doing so—discussing current challenges and obstacles for this area of research and these technologies, and projecting possibilities for the future direction of AI technology in CVD—would be of significant value for this review. If the authors chose not to do so, they should revise their abstract to more accurately reflect the core content and purpose of the manuscript.

In both the abstract and conclusion, the authors assert that “the accuracy, generalization and interpretability of the model need to be further improved.” While I do not necessarily disagree with this statement, there is little information presented in the review manuscript to justify this broad, sweeping claim. Either this assertion should be better justified throughout the manuscript, or the conclusion should be revised to reflect the information presented within the text.

Author Response

  We appreciate the reviewer for the comments regarding the potential of this manuscript to be an interesting review, and have tried our best to give a point-by-point response to the constructive comment. Please see the attachment.

Reviewer 2 Report

Well organized and presented review article on the role of ML and AI to predict CVD. 

The only minor comment is that a few references from leaders in the field are missing especially on the role of shear stress and CFD to predict CVD. Also the combination of imaging and non-imaging markers to predict coronary stenosis is missing. 

The part of AI is also missing a major part which is its explainability. Discussion about explainability of AI models must be presented. 

Author Response

(The authors gave the same response as above.)

Reviewer 3 Report

The authors provide valuable information on the interplay between state-of-the-art biomechanical and machine-learning CV modelling. However, there are some issues regarding the message and the conclusion of this review.
- The authors describe the scope of the review as: "In this review, we systematically summarized the recent research progress of artificial intelligence applications in cardiovascular diseases, including the prediction of morbidity or mortality from cardiovascular disease and the prediction of cardiovascular biomechanics modeling."
  In this context, Section 3.1 is quite brief and omits relevant ML research for many types of CV disease that have been published so far. The search criteria and the rationale for their choice are not clearly described.
- Section 3.1 implies the better predictive performance of ML models in comparison with traditional or linear models. To derive the insight, the section should provide an approximate scale of improvement in the predictive performance of ML models over "traditional methods".
- This review does not clearly describe the relation between the direct modelling of CV risk from input features (Section 3.1) and biomechanical modelling (Section 3.2).  Therefore,  the implications of the two approaches and the message of the review are not clear.
- Authors wrote in the introduction: "We summarized several common difficulties of AI technology and prospected the future development of AI technology in cardiovascular disease". However, a clear summary of future steps is missing. The summary is limited to "the accuracy, generalization and interpretability of the model need to be further improved". Could the authors elaborate more on this issue?

Author Response

(The authors gave the same response as above.)

Round 2

Reviewer 1 Report

I appreciate the attention and consideration the authors have given to my recommendations and concerns. The authors have made substantial and constructive revisions to their manuscript. I particularly appreciate the efforts made in improving and adding tables summarizing the methods and articles reviewed in the key sections of the manuscript (Tables 1 and 2) and the discussion on the challenges and potential of the field (“Challenges and Future Prospects”)—these add significant value to the review. However, I believe there remain opportunities for the authors to improve the clarity and organization of their article.

In the new paragraph on deriving complex shapes of atherosclerotic lesions from intravascular imaging, there are some statements that are either unclear or inaccurate. For example, angiography and MRI are forms of vascular imaging, but not intravascular imaging, as misstated in the opening sentence of that paragraph (lines 311-313). Additionally, the example cited for IVOCT ([110], line 313) does not actually pertain to IVOCT at all, but rather an entirely different type of imaging modality (ultrasound). Consequently, none of the works leveraging IVOCT to construct patient-specific models of diseased (e.g. atherosclerotic) arteries are included in this review.

I also believe Table 2 would benefit from being more closely aligned with information in Table 1, for example by reporting input data type.

Broadly, I still believe that Section 3 would benefit from additional context, detail, and descriptions of the individual works that would help readers to better understand the described works, their relevance, and their relationships. The section remains rather disjointed and without a clear and coherent framework or continuity in the description of the reviewed field.

If given the opportunity, I hope the authors will strive to address these persisting and new concerns.

Author Response

We again appreciate the reviewer for the valuable comments, and we are honored to have the opportunity to revise the manuscript based on these suggestions. Please see the attachment.

Reviewer 3 Report

No further comments

Author Response

We again appreciate the reviewer for the valuable comments, and we refined the English expression with the help of a native English writer.